# Immune Dysregulation in HFpEF: A Target for Mesenchymal Stem/Stromal Cell Therapy

**DOI:** 10.3390/jcm9010241

**Published:** 2020-01-16

**Authors:** Ruxandra I. Sava, Carl J. Pepine, Keith L. March

**Affiliations:** 1Center for Regenerative Medicine, University of Florida, Gainesville, FL 32610, USA; keith.march@medicine.ufl.edu; 2Cardiology Department, Elias Emergency University Hospital, 011461 Bucharest, Romania; 3Division of Cardiovascular Medicine, Department of Medicine, University of Florida, Gainesville, FL 32610, USA; carl.pepine@medicine.ufl.edu

**Keywords:** HFpEF, immune dysregulation, inflammation, mesenchymal stromal/stem cells, MSC, CHIP

## Abstract

Over 26 million people worldwide suffer from heart failure, a disease associated with a 1 year mortality rate of 22%. Half of these patients present heart failure with preserved ejection fraction (HFpEF), for which there is no available therapy to improve prognosis. HFpEF is strongly associated with aging, inflammation, and comorbid burden, which are thought to play causal roles in disease development. Mesenchymal stromal/stem cells (MSCs) have potent immunomodulatory actions and promote tissue healing, thus representing an attractive therapeutic option in HFpEF. In this review, we summarize recent data suggesting that a two-hit model of immune dysregulation lies at the heart of the HFpEF. A first hit is represented by genetic mutations associated with clonal hematopoiesis of indeterminate potential (CHIP), which skew immune cells toward a pro-inflammatory phenotype, are associated with HFpEF development in animal models, and with immune dysregulation and risk of HF hospitalization in patients. A second hit is induced by cardiovascular risk factors, which cause subclinical cardiac dysfunction and production of danger signals. In mice, these attract proinflammatory macrophages, Th1 and Th17 cells into the myocardium, where they are required for the development of HFpEF. MSCs have been shown to reduce the pro-inflammatory activity of immune cell types involved in murine HFpEF in vitro, and to reduce myocardial fibrosis and improve diastolic function in vivo, thus they may efficiently target immune dysregulation in HFpEF and stop disease progression.

## 1. Introduction

Heart failure (HF) affects over 26 million people worldwide, as a global pandemic in industrialized and developing countries alike [1]. The 1 year mortality rate for HF patients hospitalized in the United States has been estimated at 22%, which is higher than that of many forms of cancer [2]. Although medical care has improved the prognosis of HF with reduced ejection fraction (HFrEF), it is recognized that ~50% of HF patients in the USA have diastolic dysfunction with a preserved ejection fraction (HFpEF) [3,4]. As HFpEF is more closely associated with advanced age than HFrEF, projections show its prevalence is rising by 1% per year relative to that of HFrEF [5]. Indeed, the lack of improvement of HFpEF therapy is in stark contrast with the severity of the disease, with no available evidence-based therapy to improve prognosis [6]. The mainstay of HFpEF management is blood pressure control and diuretic treatment, with overall poor control of symptoms. With the exception of aldosterone receptor antagonists, which are included in the 2017 ACC/AHA Heart Failure Guideline update as a class IIb recommendation with the purpose of decreasing hospitalization [7], no other medication currently available decreases re-hospitalization or mortality rates in these patients. HFpEF pathophysiology remains elusive, and the heterogeneity of disease [8] together with its predilection for development in patients with multiple comorbidities, have represented major challenges for the development of clinical trials.

The clinical syndrome of HFpEF is characterized by diastolic dysfunction, leading to progressive worsening of systemic and pulmonary congestion, dyspnea and fatigability; pathology specimens and biopsies demonstrate that cardiac hypertrophy, fibrosis, vascular rarefaction and inflammatory infiltrates underlie the HFpEF syndrome [9,10]. Moreover, epidemiological data show a strong association between systemic inflammation, aging, and HFpEF [11,12]. The prevalent theory that reconciles systemic inflammation and HFpEF development focuses on the endothelium. Cardiovascular risk factors increase systemic inflammation by inducing endothelial dysfunction, reduced nitric oxide bioavailability and inhibition of protein-kinase G (PKG) signaling, which in turn leads to myocardial fibrosis, stiffening and hypertrophy [13]. This theory was investigated by the PARAGON-HF (Prospective Comparison of ARNI With ARB Global Outcomes in HF With Preserved Ejection Fraction) trial, using a combination of valsartan and the neprilysin inhibitor sacubitril, which increases PKG levels by enhancing availability of brain natriuretic peptide (BNP). Although sacubitril-valsartan did not reduce the rate of hospitalization for HFpEF and total cardiovascular deaths, signals of efficacy arose from subgroup analysis, suggesting that PKG modulation may be effective in women and patients with a lower ejection fraction (EF) [14].

A growing body of recent evidence from murine and clinical studies points toward the critical importance of a two-hit model of immune dysregulation as a key factor leading to HFpEF development (see graphical abstract). In this view, recently uncovered genetic factors and acquired risk factors collaborate to induce a heightened chronic pro-inflammatory tone and multi-organ subclinical dysfunction, respectively, thus defining a HFpEF predisposition. In the premalignant syndrome of clonal hematopoiesis of indeterminate potential (CHIP), inactivating mutations in genes controlling hematopoietic stem cell turnover and differentiation leads to enhanced proliferation of myeloid cells with pro-inflammatory properties, and to an increased T helper 17 (Th17) / regulatory T cell (Treg) ratio [15] (first hit). Furthermore, pro-inflammatory circulating immune cells are recruited into the heart by danger signals produced in the myocardium in response to traditional risk factors such as elevated BP, aortic valve stenosis, smoking and/or aging (second hit), with endothelial activation facilitating the immune cell influx [9,16,17,18,19,20]. Once infiltrated into the myocardial interstitium, monocytes become activated and aid in the recruitment and activation of Th1 and Th17 cells, via chemotactic molecules and antigen presentation [16,18,21], respectively. Together, monocytes, Th1 and Th17 cells orchestrate a chronic pro-inflammatory and pro-fibrotic response to cardiovascular risk factors, leading to pathological cardiac hypertrophy, myocardial fibrosis and diastolic impairment. Thus, HFpEF is associated with hallmarks of Th1/Th17-cell mediated autoimmune diseases, such as multiple sclerosis [22,23] and type I diabetes mellitus [24,25]. This overarching theory reconciles the epidemiological associations between systemic inflammation and aging and provides a genetic basis that explains why not all aging, hypertensive patients develop HFpEF. It also suggests that the absence of approaches particularly targeting immune dysregulation may explain why most therapies effective in HFrEF have failed to prevent HFpEF progression.

Owing to their remarkable results in targeting inflammation in autoimmune disease, interleukin blocking agents have been tested for the treatment of HF. The ATTACH (Anti-TNF Therapy Against Congestive Heart Failure) (*n* = 150) [26] and RENEWAL (Randomized Etanercept Worldwide Evaluation) (*n* = 2048) [27] trials investigated TNF-α antagonism in moderate-to-severe HFrEF. Neither trial reported improved symptoms or decreased death or heart failure hospitalization rates, with the larger dose of infliximab tested in ATTACH being associated with increased mortality. IL-1 blocking has been attempted in HFpEF. While patients treated with anakinra showed improved aerobic exercise capacity vs. placebo in the pilot D-HART study [28], these findings were not replicated in the phase II follow-up study [29]. These negative results may be explained by the specific characteristics of the inflammatory cascade that contributes to HF pathophysiology, such as [30,31]: (1) a chronic, low grade systemic inflammation, that is induced by multiple mediators besides TNF-α and IL-1β, such as damage-associated molecular patterns (DAMP) and mitochondria injury, which arise in the setting of myocardial dysfunction; (2) promotion of cell survival and beneficial tissue remodeling by low levels of TNF-α; (3) a by-stander, as opposed to a pathogenic role, of increased cytokine levels seen in HF. Thus, approaches that simultaneously focus on immunomodulation of abnormal responses and stimulation of tissue repair may offer more therapeutic promise than immunosuppression in HF.

Mesenchymal stem/stromal cells (MSCs) are multipotent stromal cells distributed throughout the body, typically on the abluminal aspect of both the microvasculature and the adventitia of larger vessels [32,33,34,35] found in tissues such as bone marrow, fat or the umbilical cord. MSCs act as sentinels and generals of tissue homeostasis, sensing cues from the surrounding environment and coordinating the response to injury, by regulating immune responses [35], cell survival, and tissue and vascular repair [36,37,38,39,40,41]. As pointed out in the editorial to the Special Issue “Mesenchymal Stem/Stromal Cells in Immunity and Disease” [35], these activities are largely mediated by paracrine factors, and are independent of MSC incorporation into host tissue [42,43,44,45]. Thus, MSCs may target both systemic inflammation and the cardiac pathological changes associated with HFpEF. However, MSC therapy in cardiovascular disease remains of unclear utility, owing to the limited results from relatively small clinical trials, as well as an incomplete understanding of their mechanism of action [46,47].

In this review, we will set the stage for future clinical trials investigating the usefulness of MSC therapy in HFpEF, by summarizing the latest research regarding immune cell involvement in HFpEF, as well as current evidence regarding the mechanisms of action of MSCs.

## 2. Importance of Immune Dysregulation in HFpEF—Key Lines of Evidence

### 2.1. Animal Models of HFpEF

A large number of animal models have been employed to study HFpEF. A detailed discussion of these is beyond the scope of this review, and the authors direct the reader to an excellent updated overview of these models [48]. Briefly, here we will discuss evidence obtained using two mice models of hypertension-induced HFpEF, provoked by salty drinking water, unilateral nephrectomy, and chronic exposure to aldosterone (SAUNA) for 30 days [9], or by continuous angiotensin II (AT II) infusions administered via mini-osmotic pumps for 4 weeks [49]. A third model recapitulates HFpEF associated with aging, as older and senescent C57BL/6 mice present with mild left ventricular hypertrophy (LVH), interstitial fibrosis and diastolic dysfunction, but without hypertension (HTN) [9]. Finally, transverse aortic constriction (TAC) induces pressure-overload LV remodeling, which is the most frequently used HFpEF animal model. While the first three models adequately represent the pathologic changes that occur in human HFpEF, the TAC model does not [48]. Following TAC, animals initially develop compensated LVH, followed by progression to HFpEF, with LVH, diastolic dysfunction and lung edema. However, mice further progress to HFrEF, with systolic dysfunction and cardiac dilation, which is uncharacteristic of the evolution of human HFpEF [48]. Thus, TAC, although being the most commonly used experimental model for HFpEF, more closely mirrors the cardiac pathology that ensues from aortic stenosis [21,48].

### 2.2. The Interplay of CHIP, Immune Dysregulation, and Cardiovascular Disease

Clonal hematopoiesis of indeterminate potential (CHIP) is a pre-malignant state characterized by expansion of hematopoietic stem cell (HSC) clones harboring mutations in genes encoding for epigenetic regulators of hematopoiesis, in the absence of other hematological abnormalities [50]. Most frequently, the DNMT3A, TET2, ASXL1 and JAK2 genes are affected by inactivating mutations. Mutated clones retain the capacity to differentiate into mature monocytes, granulocytes and lymphocytes, and can thus be detected in peripheral blood. CHIP is strongly associated with aging, with CHIP-associated mutations being rare in individuals younger than 40, but found in at least 10% of those aged 70 years or above [51]. As CVD is also linked to aging, the question arose whether there was any relationship between CHIP and CVD. In their seminal paper, Jaiswal et al. analyzed whole exome sequencing from 4726 patients with coronary heart disease (CHD) and 3529 controls and found that possessing a CHIP mutation imparted a 2-fold increase in the risk of incident CHD, and a 4-fold increase in the risk of early-onset myocardial infarction [51]. Murine experiments demonstrated that mice with TET2-deficient HSC had larger atherosclerotic plaques [51,52]. This association appeared to be causal, as TET-2 inactivation in macrophages enhanced their pro-inflammatory activity upon antigen stimulation, with increased production of pro-inflammatory molecules IL-1β [49,51,52,53], IL-6 [49,51,52,53], as well as that of single C-X-C motif (CXC) chemokines CXCL1 [51] (a neutrophil chemotactic factor), CXCL2 [51] (also known as macrophage inflammatory protein 2-alpha), CXCL3 [51] (also known as macrophage inflammatory protein-2-beta) and platelet factor 4 [51] (involved in coagulation). TET2 regulates the transcription of pro-inflammatory factors such as IL-6 by binding to their gene promoters; once bound, TET2 recruits histone deacetylase (HDAC2) at this level, thereby facilitating histone deacetylation that terminates gene transcription [54]. Moreover, TET2-deficient macrophages enhance IL-1β production by two mechanisms: (1) augmented gene transcription resulting from decreased HDAC activity, and (2) enhanced activation of pro-IL-1β, via increased priming of the NLPR3-inflammasome [52].

Another question focused on the role of TET2, and other CHIP-related mutations, in the development of heart failure (HF). To test this hypothesis, Sano et al. evaluated two animal models of HFpEF, induced by AT II infusion [49] or TAC [53]. Mice transplanted with bone marrow derived from TET2 knockout (KO) mice developed a HFpEF phenotype, with cardiac hypertrophy, fibrosis, and increased cardiac macrophage infiltration [49,53]. Deficiency of other genes associated with CHIP, such as DNMT3 deficiency, were also shown to support development of HFpEF after AT II infusion [49]. Thus, CHIP-related mutations predispose mice exposed to chronic hypertension or aortic stenosis to develop HFpEF.

CHIP-related mutations can also impair the function of T cells. Clinical data reported by Zeiher et al. showed that, among a population of patients with degenerative aortic stenosis undergoing transcatheter aortic valve implantation (TAVI), those carrying a DNMT3A mutation had an increased Th17/Treg ratio [15]. This was not observed in patients harboring a TET2 mutation. However, in murine models of disease, combined TET2 and TET3 deficiency induced T cell differentiation primarily towards a Th17 phenotype [55]. With Th17 cells being involved in auto-reactivity against self-antigens [56,57], and Treg cells being chiefly involved in maintaining tolerance to self [57], these findings support the notion that CHIP-related multiple mutations can skew immune cells towards a pro-inflammatory phenotype, that in turn supports chronic inflammation and heart failure development and progression. Indeed, in patients with ischemia-driven congestive HF, inactivating mutations in DNMT3A or TET2 were independently associated with an approximately doubled risk of death and HF hospitalization [58]. Further studies are needed to investigate this relationship in HFpEF patients, and also to determine whether other genetic mutations, unrelated to CHIP and HSC renewal, can induce pro-inflammatory immune cell phenotypes and promote HFpEF development in the context of aging and comorbid diseases.

### 2.3. Monocytes: Effectors of Cardiac Remodeling for Better or Worse

To better understand monocyte dynamics in HF, a distinction must be made among key myocardial monocyte populations, including the distinction between tissue-resident and hematogenous infiltrating macrophages. In the murine heart, resident cardiac macrophages are CCR2-, originate from embryonic sources, mostly seed the heart before birth, and contribute to tissue homeostasis, regulating angiogenesis and other repair processes [59]. In contrast, infiltrating cardiac macrophages, distinguished by CCR2 positivity, are derived from circulating monocytes and play mainly pro-inflammatory roles. While they represent a minority of cardiac macrophages in steady state conditions, stress induces a rapid increase in CCR2+ macrophage influx into the heart [59].

Myocardial stress, such as pressure overload or salt-induced hypertension, drives upregulation of CCR2 ligands, such as monocyte chemotactic protein-1 (MCP-1 or CCL2) [9,16], MCP-3 (or CCL7) [16], CCL12 [16], and SDF-1alpha [9], which play an essential role in CCR2+ monocyte influx into the heart [9,16]. Accordingly, in response to hypertension [9,59] or pressure overload [16], HSC and progenitor cells expand in the bone marrow and generate peripheral leukocytosis, with pro-inflammatory (Ly6C^++^CCR2^+^ in mice [16]) monocytes infiltrating the heart in a CCR2-dependent manner [9,16]. Clinical data collected on a small set of HFpEF patients mirror the murine findings, with enhanced monocytosis and myocardial macrophage density [9].

Macrophages can induce pathological changes within the myocardium directly, by virtue of their secreted molecules, or indirectly, by antigen presentation to T cells. Macrophages isolated by fluorescence-activated cell sorting from myocardial samples of SAUNA-exposed mice are potent producers of IL-10, which in turn induces secretion of osteopontin (OPN) in autocrine fashion. In the SAUNA model, the direct action of OPN and TGF-β, but not IL-10, induced fibroblast activation, as fibroblasts lack the receptor complex for IL-10 [9]. Following TAC, early (3–7 days following TAC) myocardial macrophage expansion was essential for later T-cell expansion, since inhibiting monocyte myocardial influx by CCR2-blockers abrogated the increase in activated T cells in mediastinal lymph nodes at 1 week. Early macrophage expansion was also critical for HF development, with antagonism of CCR2 associated with early attenuation of LVH development, together with improved systolic function and reduced LVH and myocardial fibrosis at 4 weeks [16]. However, as TAC induces an acute pressure overload, these findings cannot be readily extended to HFpEF induced by chronic stimuli, such as aging and hypertension, and further studies are required to delineate the dynamics of myocardial accumulation of macrophages in more appropriate animal models.

Clinical data also support a pathogenic role for macrophages in HFpEF development. Using peripheral blood samples from patients with HFpEF defined by stringent criteria, including a history of hospitalization for decompensated HF in the presence of normal ejection fraction, Glezeva et al. [60] investigated the effect of patient-specific serum on monocytes derived from healthy donors, as well as the surface receptors expressed by patient circulating monocytes. By measuring the level of secreted cytokines by monocytes co-cultured with patient serum, the investigators demonstrated that HFpEF patients presented a pro-inflammatory (increased TNF-α and MCP-1), pro-fibrotic [61,62] (increased CCL17), and anti-angiogenic [63] (increased CXCL10, IL-12) milieu, when compared to asymptomatic patients presenting with hypertension or diastolic dysfunction [60]. These findings were supported by evaluation of patient circulating monocytes. Classic, pro-inflammatory CD14^++^CD16^−^ monocytes were increased in both patients with asymptomatic diastolic dysfunction and those with HFpEF. However, the alternatively-activated CD14^++^CD16^+^ monocytes, which mediate resolution of inflammation, tissue repair, and fibrosis [64], were increased in HFpEF patients only [60]. We hypothesize that HFpEF involves an inadequate resolution of inflammation, with chronic inflammation and sustained tissue injury leading to ongoing activity of anti-inflammatory yet pro-fibrotic macrophage subtypes. While HF-decompensating factors such as hemodynamic overload or uncontrolled hypertension [65] or diabetes [66] can elicit tissue damage, as demonstrated by baseline/small elevations in cardiac troponins, HFpEF still progresses in spite of adequate risk factor control [67]. We further hypothesize that activation of innate immune cells leads to recruitment and activation of T cells, which in concert support the development of a pro-inflammatory and pro-fibrotic environment in the myocardium.

### 2.4. T Cells as Mediators of Chronic Heart Failure Development

T cells are major components of the adaptive immune system that have only recently been implicated in the pathogenesis of heart failure induced by non-infectious stimuli. While this paper will only briefly touch upon current evidence regarding the role of T cells in HFpEF, we refer the reader to an excellent in-depth review on the topic of T-cell involvement in HF [57]. In parallel with their increased specificity of action, T helper cells are phenotypically heterogeneous. Briefly, lymphoid precursors originating in the bone marrow populate the thymus, where they differentiate into naïve CD4^+^ and CD8^+^ cells, and into natural T regulatory (Treg) cells. Naïve CD4^+^ cells then migrate to secondary lymph nodes, where they develop into four main types of T cells: T helper (Th) 1, Th2, Th17 and Tregs, which produce their signature cytokines, IFN-γ, IL-4, IL-17, and TGF-β and IL-10, respectively [57].

Unfortunately, there is a scarcity of data regarding the role of T cells in “true” animal models of HFpEF [48], with most experiments being conducted using the pressure-overload model. Although the elevation of pressure in these models is acute, and not chronic, valuable hypothesis-generating information regarding the impact of T cells in cardiac fibrosis and hypertrophy can be derived from these experiments. T cell recruitment into the heart requires a chemotactic gradient, together with expression of selectins and integrins on the cardiac endothelium. AT-II infusions were associated with heightened expression of ICAM-1 and VCAM in murine hearts [17]. Following TAC, T cell chemotactic factors [18,68,69] CXCL10 [21,70] and CXCL11 [21], CCL2 [16,21] or CCL5 [21] are produced by stressed cardiomyocytes, cardiac fibroblasts and CCR2^+^ macrophages. The CXCR3–CXCL9/CXCL10/CXCL11 axis [18] appears to play a major role in T cell recruitment into the heart, as CXCR3^−/−^ mice displayed reduced T cell, but intact macrophage infiltration. Furthermore, CXCL9 and CXCL10 were shown to enhance adhesion of T cells to ICAM-1 in a LFA-1 dependent manner [19]. Interestingly, there is evidence to support the involvement of the CXCR3–CXCL10 axis in human HF as well, since higher circulating levels of CXCL10 [71] and increased myocardial infiltration of CXCR3^+^ cells [19] have been described in end-stage HFrEF patients. While these mechanisms require further study in HFpEF, cardiac biopsies obtained from such patients show increased ICAM-1 expression [20], demonstrating the presence of part of the mechanism required for attracting T cells into myocardium.

CD4^+^ T cell infiltration has been shown in murine hearts following 2 weeks of AT-II infusion, in turn associated with cardiac hypertrophy and fibrosis. Th1 cells appear to be partially responsible for the findings, as these were all attenuated (albeit not abrogated) in IFN-gamma^−/−^ mice [17]. Moreover, T cell infiltration was also associated with, and required for, macrophage accumulation [17]. This supports a positive feedback loop between macrophages and T cells, with macrophages initially recruiting T cells in response to myocardial damage, followed by T cells supporting chronic myocardial macrophage infiltration. Although the specific antigen is not known, this feedback loop appears to depend on antigen presentation, as blocking co-stimulation of T cells by abatacept led to decreased macrophage activation and infiltration into the heart [21]. Conversely, Tregs may be protective against development of diastolic dysfunction, since specific reconstitution of Treg cells attenuated myocardial fibrosis [17,72]. In clinical settings, the Th17/Treg cell ratio was increased in patients carrying the CHIP-causing DNMT3A mutation and afflicted by severe aortic stenosis, which is associated with a HFpEF-like syndrome. Such an association has also been reported in HFrEF and HFpEF patients. However, it is uncertain whether the cells labeled by the authors as Tregs based on a CD4^+^/ CD25^+^ phenotype selected solely Treg cells, since these markers are also expressed on effector T cells [73]. While FOXP3 is the preferred marker for natural Tregs, it is not consistently expressed on induced Tregs; instead, the expression of CD127 can be used in addition to CD4 and CD25 to differentiate induced Tregs (CD127^−^) from effector T cells [73]. Further research needs to discern whether Treg cells contribute to or are merely associated with the pathogenesis of HFpEF.

Following TAC, increased numbers of T cells could be detected in the mediastinal lymph nodes and myocardium as early as 2 and 7 days, respectively [21]. They were also present in the chronic phase of HF, at 4 [16,21,74,75] and 6 [70] weeks after TAC. The vast majority of these cells appear to be CD4^+^ IFN-γ producing Th1 cells [75]. Their presence was associated with increased expression of adhesion molecules [74], production of pro-inflammatory mediators IL-1β, TNFα and IL-6 [21,74], cardiac fibrosis [21,70,74,75] and hypertrophy [21,70,74,75]. In various models of genetic or post-natal T cell depletion, CD4^+^ Th1 producing cells appear to be necessary for the development of cardiac fibrosis [70,75]. Indeed, IFN-γ producing T cells obtained from the mediastinal lymph nodes of mice subjected to TAC mediate fibrosis by direct contact, inducing fibroblast to myofibroblast differentiation by binding to cardiac fibroblasts via integrin alpha-4, in an IFN-γ dependent manner [75]. Evidence regarding the involvement of T cells in TAC-induced cardiac hypertrophy is less clear and requires further investigation [70,74]. When considering experimental models, post-TAC depletion strategies are probably favorable to genetic T cell deletion, as the former more closely mimics a therapeutic approach, while results seen with the latter may be confounded by the influence of T cells on the development of normal immune responses [74]. Finally, T cells likely represent the main source of pro-inflammatory molecules and upregulate endothelial adhesion molecules, as mice with a genetic deficiency in T cells (TCRα^−/−^) present no increase in mRNA of TNF-α, IL-1β, IL-6, E-selectin, VCAM, and ICAM following TAC. Taken together, these data suggest that the presence of Th1 cells support a pro-inflammatory myocardial milieu, which is known to be associated with hallmarks of the HFpEF syndrome [76,77] such as increased cardiomyocyte turnover [31,78] and endothelial [78] and diastolic dysfunction [29].

## 3. Complex Therapies for a Complex Disease—Exploring the Potential of Mesenchymal Stem/Stromal Cell Therapies for HFpEF

Mesenchymal stem/stromal cells (MSCs) are multipotent mesenchymal cells that can be found in tissues such as the bone-marrow, fat, the umbilical cord or placenta [79,80]. While initial enthusiasm regarding MSCs originated from their capacity to differentiate into various cell types, it is now widely believed that tissue incorporation of exogenously-administered MSCs is negligible and does not account for the therapeutic effect of MSC infusions [42,43,44,45]. Conversely, MSCs have emerged as both sentinels and generals of tissue homeostasis, which they regulate through secretion of an extensive repertoire of paracrine factors. MSCs support tissue and vascular repair, by producing growth factors such as VEGF, HGF, bFGF, angiopoietin-1 and CXCL10 [36,37]. They also promote the survival of resident cells and the proliferation and differentiation of endogenous stem cells, through epidermal growth factor (EGF), HGF, insulin-like growth factor (IGF) or stem cell-derived factor 1 (SDF-1) [36]. Most interestingly, MSCs modulate the response of immune cells via indolamine 2,3-dioxygenase (IDO), transforming growth factor-β (TGF-β), IL-10, prostaglandin E2 (PGE2), TNF-α–induced gene/protein 6 (TSG-6) and hepatocyte growth factor (HGF) [38,39]. MSCs can also “sense” the nature of inflammation, responding differently in conditions of sterile or bacterial-driven inflammation [41]. While priming MSCs with pro-inflammatory factors such as TNF-α induces a mainly suppressive MSC-phenotype, the opposite can be seen when MSCs are primed with bacterial factors such as LPS [40], which activates MSCs to promote microbial clearing [81]. As a result of these ambivalent traits, MSC infusions are not associated with increased risk of infection [82], as opposed to cytokine-blocking therapies, for which life-threatening infections represent a major side effect [83]. Thus, MSCs, as well as MSC-products such as MSC-conditioned media, have considerable appeal for the treatment of diseases dominated by pathological inflammation. Mesoblast’s allogeneic mesenchymal precursor cells (alloMPC) has been approved for the treatment of acute graft-versus-host disease in Japan, with this being among the first regulatory approvals of an allogeneic MSC therapy. In the cardiovascular field, a recent meta-analysis demonstrated that autologous cell therapy in critical limb ischemia improved ulcer healing, reduced the amputation rate and amputation-free survival [84]. While MSC therapy for HFpEF has not been attempted in a clinical context, we will summarize the in vitro and in vivo evidence that recommends MSC as a promising therapeutic agent in this disease.

### 3.1. In vitro Immunomodulatory Properties of Mesenchymal Stem/Stromal Cells

Co-cultures of MSCs and macrophages have shown that MSCs induce conversion of classically activated, pro-inflammatory macrophages (M1) to alternatively activated, pro-reparatory macrophages (M2) [39]. M2 macrophages mediate the resolution of inflammation, by phagocytosis of cellular debris, production of extracellular matrix proteins, and secretion of cytokines such as IL-10 and TGF-β [64,67,85]. IL-10 and TGF-β promote the resolution of inflammation, by suppressing the pro-inflammatory activities of antigen presenting cells [85] and supporting wound healing [64]. However, they accomplish their latter goal by supporting fibrosis development [9,86,87]. Assessing the pathophysiological impact of macrophages is further complicated by the exceptional heterogeneity and plasticity of macrophages, which can completely alter their gene expression patterns based on their environment in as little as 7 days [88]. Some authors consider the M1/M2 polarization paradigm as obsolete, suggesting that the definition of macrophage type should include information regarding their origins, environmental stimuli, and their dynamics in the course of the inflammatory process [89]. We surmise that in vivo studies are required to ascertain whether MSCs therapy induce a transient M2 phenotype that effectively clears inflammation, as opposed to persistent M2 signaling that may aggravate fibrosis in HFpEF.

A wealth of in vitro data supports the immunomodulatory properties of MSC on T cells. In co-culture models, MSCs suppress the proliferation of CD4+ T cells induced by alloantigens [90]. This is largely dependent on the production of soluble mediators such as IDO, TGF-β and PEG2 [38,91]. Modulatory mechanisms requiring cellular contact mediated by molecules such as PD-L1 [92] have also been described. However, as studies preventing cell contact did not demonstrate reduced immunomodulation of CD4^+^ T cells by MSC [93], it is possible that direct contact serves a redundant purpose. Although MSCs produce small amounts of immunomodulatory molecules under baseline conditions, full suppression of CD4^+^ T cell proliferation requires MSC “licensing” by pro-inflammatory factors [38]. Interestingly, signaling through IFN-γ, the signature cytokine produced by Th1 cells, was shown to activate the anti-proliferative effects of MSCs on CD4^+^ T cells, as administering anti-IFNγR monoclonal antibodies restored T cell proliferation [93]. Thus, MSCs may have evolved as a control mechanism against abnormal T cell responses. Besides direct effects on CD4^+^ T cells, MSCs can also increase Treg numbers by inducing their expansion [38,93] and the conversion of Th17 to a Treg phenotype [39], via TGF-β secretion [39]. Tregs block effector T cells both in a paracrine fashion, through secretion of inhibitory cytokines TGF-β and IL-10 [94], as well as by direct contact, via constitutively-expressed CTLA4, which binds to T-cell expressed CD28 thereby activating inhibitory signaling pathways [95]. Thus, MSCs may “switch off” inappropriate Th1 signaling in HFpEF through multiple pathways, all of which carry a significant in vitro pro-fibrotic potential via production of IL-10 and TGF-β. Similar to the relationship with macrophages, the relative contribution of anti-inflammatory versus pro-fibrotic signaling of MSCs in the context of HFpEF-related myocardial inflammation requires in-vivo exploration.

### 3.2. Evidence of Cell Therapy in HFpEF Animal Models

A limited number of cell therapy studies have been performed in animal models of HFpEF, and the impact on pro-inflammatory factors and immune cells has not been evaluated consistently. Early, subclinical diabetic cardiomyopathy is defined by low levels of inflammation and absence of overt cardiac pathology. After administering placenta-expanded MSC-like cells shortly after diabetes induction with streptozotocin, investigators reported reduced cardiac expression of VCAM1 and IFN-γ at 2 weeks post- cell therapy, suggesting MSC therapy was associated with reduced recruitment of Th1 lymphocytes into the myocardium [96]. Reduced infiltration of macrophages (CD68^+^) and leukocytes (CD45^+^), and reduced production of TNF-α, IL-1β and IL-6, were reported at 4 weeks after intracoronary (IC) injection of cardiosphere-derived cells (CDCs), a mixture of cardiac-origin progenitor and mesenchymal cells, in Dahl salt-sensitive mice exposed to a hypernatremic diet [97]. MSC treatment was also associated with improved diastolic dysfunction, ascertained by the normalization of the echocardiographic E/a ratio, and of the invasively measured LVEDP and Tau relaxation constant [97]. This therapeutic effect was at least partially due to reduced myocardial fibrosis, the magnitude of which was similar to fibrosis seen in control animals [97]. Although the specific mechanisms were not investigated, CDC therapy was associated with reduced fibroblast proliferation and decreased transition to myofibroblasts [97]. We hypothesize this may be a consequence of decreased Th1 influx into the heart, as IFN-γ producing cells have been shown to support fibrosis development by direct contact with cardiac fibroblasts [75]. Finally, intravenous (IV) infusion of stromal vascular fraction (SVF) was also shown to reduce diastolic dysfunction and LV collagen content at 4 weeks after treatment in an aging rat model [98]. Similar to data regarding T-cell antagonism, reports regarding the effects of MSCs on LVH are conflicting, spanning from minimal [98] to no effect [97].

### 3.3. Immunomodulatory Effects of MSCs in Preclinical Models of Th1-Mediated Diseases

While the modulatory effects of MSC therapies on T cell phenotypes in HFpEF has not been studied in detail, clues can be inferred from animal models of other Th1-mediated diseases, such as type I diabetes mellitus (T1DM) and multiple sclerosis (MS) [99]. In two different mouse models of T1DM, IV infusion of MSC [100,101] or MSC-derived extracellular vesicles [100] (MSC-EV) was associated with induction of tolerogenic APCs [100,101], with increased secretion of IL-10 [100,101] and TGF-β [101], and reduced secretion of Th1-(IFN-γ [100], IL-12p35 [101] and IL-12p70 [101]) and Th17- (IL-6 [100] and IL-17A [100]) related cytokines. This was associated with less inflammatory infiltrates [100,101], but more FoxP3^+^ Treg cells [101], in pancreatic islets from diabetic mice treated with IV MSCs. Furthermore, MSCs also suppressed Th1 polarization directly, via secretion of TSG-6 [101]. Immunomodulation of the Th1–Th17 axis by MSC therapy has also been reported in numerous animal models of MS, which accompanied reduced expression of pro-inflammatory cytokines in the brain [102,103], diminished inflammatory infiltrates in the brain [102,104] and spinal cord [105], and improved motor function [102,105]. Interestingly, some of these studies suggest MSC-induced immunomodulation can cross species, as IV administration of EVs derived from human MSC lowered plasmatic levels of Th1- and Th17-related cytokines, and improved clinical outcomes [104,105].

### 3.4. Immunomodulatory Effects of MSCs in Clinical Trials of Th1-Mediated Diseases

Recognition of potent in vitro and in vivo MSC-mediated modulation of Th1 cell proliferation and activation has led to clinical trials investigating the therapeutic utility of MSC infusions in autoimmune diseases dominated by autoreactive Th1 and Th17 responses. Multiple small studies support that various MSC strategies can improve disease course in patients with established T1DM. Patients receiving a single IV infusion of autologous, bone-marrow derived MSCs, had preserved levels of C-peptide peak values and C-peptide AUC following a mixed-meal tolerance test at one year after treatment, while control patients exhibited reductions in both parameters [106]. In another study, a single injection of a mixture of allogeneic umbilical-cord MSCs together with autologous, bone-marrow derived mononuclear cells into the pancreatic artery led to increased C-peptide area under curve (AUC) at 12 months after therapy [107]. Although these studies did not investigate the mechanism underlying the clinical improvement, other studies using related, albeit different cell types, support it is dependent on MSC immunomodulatory properties. Patients treated by Stem Cell Educator Therapy, a device that separates patient’s lymphocytes and co-cultures them with cord blood-derived multipotent stem cells, which contain a mixture of MSCs and other cells, exhibited increased circulating FOXP3+ Treg cells 4 weeks later [108]. Furthermore, in patients with newly-developed T1DM, IV administration of autologous HSC, which are stem cells of a lineage different than that of MSCs, was associated with reduced Th1 and Th17 activation [109]. Specifically, analysis of PBMCs collected from patients 12 months after stem-cell educator therapy demonstrated reduced numbers of circulating Th1 and Th17 cells, together with diminished production of their transcription factors, Tbet and ROR-γt, and of their signature cytokines, IFN-γ and IL-17A. Conversely, Treg numbers were increased [109].

Beneficial effects of MSC therapy were also reported in a small-scale clinical trial that enrolled MS patients with refractory disease. MSC treatment was associated with a trend toward reduced plaque formation [110] and prevention of disease progression [111], together with a statistically insignificant lowering of circulating Th1 cell numbers [110]. Based on these findings, the MEsenchymal StEm cells for Multiple Sclerosis (MESEMS) Network will conduct a multi-national, randomized, double blind, cross-over, phase I/II clinical trial with the primary objective of demonstrating safety and efficacy of autologous, bone-marrow derived MSC therapy vs. placebo in MS [112].

### 3.5. MSC Clinical Trials in HF—Lost in Translation?

No clinical trial to date has tested the effectiveness of MSC therapy in HFpEF. However, multiple small MSC trials in HFrEF have been conducted, and thus far have yielded heterogeneous results, possibly as a result of insufficient understanding of MSC mechanisms of action and consequent inclusion of patients unlikely to benefit from therapy, the enrolment of heterogeneous patient populations, the lack of standardized cell-processing protocols, and differences in study design [113,114]. We and others [113] believe an absence of therapeutic effect is unlikely, as MSCs have been associated with a ~7%–9% increase in LVEF after acute myocardial infarction at 4–6 months after cell therapy. The lack of persistence of effect is, in our view, entirely expectable after a single cell administration, and sustained therapeutic benefit will most likely require reinfusion of MSCs at later time points. Moreover, both autologous [115] and allogeneic [115,116,117] MSC delivery through intravenous [117], intracoronary [118] or transendocardial [116] routes have proven universally safe in patients with HFrEF, and metanalyses have demonstrated statistically significant, albeit small improvement in many of the variables evaluated [119]. Moreover, some of these trials also support the notion that MSC therapy can induce immunomodulation in the setting of chronic HF. Transendocardial administration of autologous or allogeneic MSCs in nonischemic dilated cardiomyopathy (NIDCM) were both associated with immunomodulatory effects six months after initial therapy, with decreased levels of circulating TNF-α [115]. This was accompanied by a lower circulating number of early (CD3^+^ CD69^+^) and late (CD3^+^ CD25^+^) activated T cells [115]. However, in a separate study, IV infusion of allogeneic MSCs in NIDCM patients was surprisingly associated with increased circulating total T cells (CD3^+^) and T helper (CD4^+^) cells at 30 days post-infusion—an effect that disappeared at 3 months [117]. Measurement of heterogenous cell populations, such as total T cells or all T helper cells, as opposed to T cell subtypes, hinders the interpretation of this data. Additionally, future studies investigating the immunomodulatory properties of MSC in HFpEF should include evaluation of immune cells shown to be involved in HFpEF pathogenesis in animal models, such as classical (CD14^++^CD16^−^) [120] and alternatively-activated macrophages (CD14^+^ CD16^++^) [120], and T cell subsets Th1 (CD3^+^, CD4^+^, expression of the T-bet transcription factor, and production of IFN-γ) [57], Th17 (CD3^+^, CD4^+^, expression of ROR-γt transcription factor, and production of IL-17) [57], natural Tregs (CD3^+^ CD4^+^ CD25^+^ FoxP3^+^, production of TGF-β, inhibition of effector Th cells [57,73,121], peripherally-induced Treg (CD3^+^ CD4^+^ CD25^+^ CD12^−^, production of TGF-β, inhibition of effector Th cells) [57,73,121] for peripherally-induced, with the chosen markers being specific enough to ensure the identity of the cells under investigation with a reasonable degree of certainty. Furthermore, evaluation of immunomodulatory effects should be performed at multiple time points, in order to better define the dynamic process of inflammation, and to detect persistence of response to cell therapy.

The therapeutic effect of MSC cell-based therapy on diastolic dysfunction has not been investigated. However, both CD34^+^ [122] and bone marrow-derived mononuclear cell (BM-MNC) therapy [118,123,124] have been associated with improved diastolic function in NIDCM and chronic ischemic cardiomyopathy, respectively. The mechanisms of action of these cell preparations are poorly understood. The beneficial effect of autologous CD34^+^ cells appears to be mediated at least partly by vascular repair [125,126]. BM-MNC include stem cell populations such as HSCs, MSCs, and endothelial progenitor cells, which may facilitate cell survival and tissue repair [113]. Multiple preclinical and clinical studies comparing MSCs and BM-MNCs suggest MSCs commonly have superior therapeutic effects. In a murine model of chronic myocardial ischemia, MSC therapy was associated with enhanced angiogenesis and smaller collagen scars than BM-MNC therapy, and only the former improved cellular metabolism [127]. In a clinical trial enrolling patients with critical limb ischemia, MSCs were shown to possess superior wound healing and angiogenesis properties, being associated with reduced pain and improved limb perfusion when compared to BM-MNC therapy [128]. Thus, MSCs may be more effective inducers of reverse cardiac remodeling in HFpEF than BM-MNC.

## 4. Conclusions

Multiple lines of evidence suggest that dysregulation of inflammation and/or immunity underlie the pathology and progression of various phenotypes of HFpEF. Genetic mutations associated with CHIP induce skewing of immune cells towards a pro-inflammatory phenotype in vitro and in vivo, supporting the development of HFpEF in mice exposed to chronic hypertension. In preclinical models, cardiovascular risk factors lead to cardiac infiltration and activation of monocytes, Th1 and possibly Th17 cells, which support endothelial dysfunction and directly induce myocardial fibrosis. Investigation of these pathways in the clinic are in their infancy; however, immune dysregulation has been reported in patients with CHIP mutations and severe aortic stenosis, and CHIP-related mutations are associated with a marked increase in HF hospitalizations in patients with chronic ischemia. Moreover, the fact that neither BP nor risk factor control can stop the progression of HFpEF strongly argue in favor of an important pathophysiological mechanism which remains unopposed by current therapeutic approaches. MSCs are key regulators of tissue homeostasis, and both in vitro and in vivo studies suggest they may combat systemic inflammation as well as the excessive myocardial fibrosis and vascular rarefaction seen in hearts of patients with HFpEF. Furthermore, numerous studies have demonstrated that MSCs are safe to administer in HF patients. Accordingly, we hypothesize that MSCs or other cell-based products, which are biologically active in modulating immunity and facilitating tissue repair, represent an intriguing therapeutic tool to evaluate in clinical trials directed to management of HFpEF.

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
