# Peer review of "Immune Dysregulation in HFpEF: A Target for Mesenchymal Stem/Stromal Cell Therapy"

_jcm, 2020, doi:10.3390/jcm9010241_

Round 1
Reviewer 1 Report
In this manuscript, Sava and co-workers suggested the roles of MSCs in enhancing the therapeutic potential of MSC in immune dysregulation at the heart of the HFpEF . The reviwer paper sounds good. I descieded to accept without any correction.
Author Response
Thank you for taking the time to review our paper, as well as for accepting it.
Reviewer 2 Report
Manuscript ID: jcm-689599
MS Title: Immune Dysregulation in HFpEF: A Target for mesenchymal Stem/Stromal Cell Therapy
Comments to the Author
This manuscript describes the role of immune system and genetic mutations on the development of HFpEF and suggest MSCs as a potential therapeutic option, due to its immunomodulation and reparative properties. The review is clear, updated and informative, pointing out important epidemiology, pathophysiology and therapeutical approaches for HFpEF.
Some minor point is suggested below:
1. Line 254: Reference 17 doesn’t match to the text
2. Line 280: Cite reference
3. Line 287: Cite reference for MSC presence on tissues, even if they were provided before
4. The section 3.4 intend the discussion of the MSC effects on immunomodulation, however the authors start talking about MSC and reference studies using Hematopoietic Stem Cell. It is confused. I'd suggest changing the name of the section to “…Stem Cell Therapy….” Instead of MSC or rewrite the initial part of the section.
5. Line 401: reference is not correct
6. Line 401: “…autologous HSC induced a long lasting immumodulatory effect…” Same problem. The authors were discussing about MSC, switched to HSC and switched back to HSC with no linearity.
7. Recheck the overall references.
Author Response
Thank you for taking the time to thoroughly review our paper. We appreciate your feedback and have made the suggested corrections and checked the overall references.
In reference to "Line 401: “…autologous HSC induced a long lasting immumodulatory effect…” Same problem. The authors were discussing about MSC, switched to HSC and switched back to HSC with no linearity. ". We have rewritten the paragraph, clearly delineating evidence obtained using MSCs, and separately mentioning the HSC study due to it's in depth analysis of the mechanism underlying cell therapy. The paragraph now reads:
"3.4. Immunomodulatory Effects of MSCs in Clinical Trials of Th1-Mediated Diseases
Recognition of potent in vitro and in vivo MSC-mediated modulation of Th1 cell proliferation and activation has led to clinical trials investigating the therapeutic utility of MSC infusions in autoimmune diseases dominated by autoreactive Th1 and Th17 responses. Multiple small studies support that various MSC strategies can improve disease course in patients with established T1DM. Patients receiving a single IV infusion of autologous, bone-marrow derived MSCs, had preserved levels of C-peptide peak values and C-peptide AUC following a mixed-meal tolerance test at one year after treatment, while control patients exhibited reductions in both parameters[105]. In another study, a single injection of a mixture of allogeneic umbilical-cord MSC together with autologous, bone-marrow derived mononuclear cells into the pancreatic artery led to increased C-peptide area under curve (AUC) at 12 months after therapy [106]. Although these studies did not investigate the mechanism underlying the clinical improvement, other studies using related, albeit different cell types, support it is dependent on MSC immunomodulatory properties. Patients treated by Stem Cell Educator Therapy, a device that separates patient’s lymphocytes and co-cultures them with cord blood-derived multipotent stem cells, which contain a mixture of MSCs and other cells, exhibited increased circulating FOXP3+ Treg cells 4 weeks later[107]. Furthermore, in patients with newly-developed T1DM, IV administration of autologous HSC, which are stem cells of a lineage different than that of MSCs, was associated with reduced Th1 and Th17 activation[108]. Specifically, analysis of PBMCs collected from patients 12 months after Stem-cell educator therapy demonstrated reduced numbers of circulating Th1 and Th17 cells, together with diminished production of their transcription factors, Tbet and ROR-γt, and of their signature cytokines, IFN-γ and IL-17A. Conversely, Treg numbers were increased[108]."
Reviewer 3 Report
This is an excellent review that draws the reader’s attention to the growing problem of heart failure with preserved ejection fraction (HFpEF). The prevalence of HFpEF is increasing and unlike HF with reduced ejection fraction (HFrEF), this disease responds poorly to pharmacological treatments such as ACE inhibitors or β blockers, thereby creating the need for alternative therapeutic strategies.
The authors make a convincing case that immune dysregulation is a major contributor to HFpEF, and provide a detailed description of the role of the immune system in cardiovascular disease. They go on to make the case that mesenchymal stem/stromal cells (MSCs), which are known to have immunomodulatory effects, could be used to combat this disease.
While this review is well-written and thorough, I have a few questions and suggestions –
The authors make a convincing case that combating uncontrolled inflammation is a reasonable therapeutic strategy, however, I feel the jump to proposing MSCs as a therapy is sudden. I think it would be instructive to first briefly discuss the use of chemical inhibitors of inflammation, and then segue to the use of MSCs (perhaps even comparing the two strategies.) Although they raise the valid concern that such inhibitors could expose patients to infections, I still think they should discuss studies that have yielded promising results (e.g. [1]). The authors briefly mention that MSC therapy for HFrEF has “yielded heterogenous results”. Considering immune dysregulation is also a feature of HFrEF, it would be useful to further expand on why MSC therapy has yielded modest results in this case. Is this the result of experimental weaknesses or the lack of potency of this therapy? Are there reasons to believe that MSC therapy would be more successful in combating HFpEF? If so, why? A simple summary figure that recapitulates the major points of this review would be useful to the reader.
Minor corrections –
Line 34 – change “patients than” to “patients more than”
Lines 51, 52 – expand “CVD”, “NO”, and “PKG”
Lines 103, 104 – expand “LVH” and “HTN”
Line 148 – expand “TAVI”
Line 165 – add a comma after “CCR2-”
Line 321 – change “later” to “latter”
Line 335 – change “possibly” to “possible”
References:
Abbate, A., et al., Interleukin-1 blockade with anakinra to prevent adverse cardiac remodeling after acute myocardial infarction (Virginia Commonwealth University Anakinra Remodeling Trial [VCU-ART] Pilot study). Am J Cardiol, 2010. 105(10): p. 1371-1377 e1.
Author Response
Thank you for your in depth review of our paper and for your valuable advice.
We have added the following paragraph on citokine blocking therapies in the introduction:
"Owing to their remarkable results in targeting inflammation in autoimmune disease, interleukin blocking agents have been tested for the treatment of HF. The ATTACH (n=150) [26] and RENEWAL (n=2048) [27] trials investigated TNF-α antagonism in moderate-to-severe HFrEF. Neither trial reported improved symptoms or decreased death or heart failure hospitalization rates, with the larger dose of infliximab tested in ATTACH being associated with increased mortality. IL-1 blocking has been attempted in HFpEF. While patients treated with anakinra showed improved aerobic exercise capacity vs. placebo in the pilot D-HART study [28], these findings were not replicated in the phase II follow-up study [29]. These negative results may be explained by the specific characteristics of the inflammatory cascade that contributes to HF pathophysiology, such as [30, 31] : 1) a chronic, low grade systemic inflammation, that is induced by multiple mediators besides TNF-α and IL-1β, such as damage-associated molecular patterns (DAMP) and mitochondria injury, which arise in the setting of myocardial dysfunction; 2) promotion of cell survival and beneficial tissue remodeling by low levels of TNF-α; 3) a by-stander, as opposed to a pathogenic role, of increased cytokine levels seen in HF. Thus approaches that simultaneously focus on immunomodulation of abnormal responses and stimulation of tissue repair may offer more therapeutic promise than immunosuppression in HF."
We have also discussed potential causes of heterogenous results of prior cell therapy trials in the following paragraph:
"However, multiple small MSC trials in HFrEF have been conducted, and thus far have yielded heterogeneous results, possibly as a result of insufficient understanding of MSC mechanisms of action and consequent inclusion of patients unlikely to benefit from therapy, the enrolment of heterogeneous patient populations, the lack of standardized cell-processing protocols, and differences in study design [112, 113]. We and others [112] believe an absence of therapeutic effect is unlikely, as MSCs have been associated with a ~7-9% increase in LVEF after acute myocardial infarction at 4-6 months after cell therapy. The lack of persistence of effect is, in our view, entirely expectable after a single cell administration, and sustained therapeutic benefit will most likely require reinfusion of MSCs at later time points. "
We have also added a summary figure to the manuscript.